# The effects of prophylactic use of paracetamol on body temperature and blood pressure in elderly patients with acute stroke: Data from the PRECIOUS trial

Jeroen C. de Jonge[1,2]*, Wouter M. Sluis[1], Hendrik Reinink[1,3], Philip Bath[4], Lisa J Woodhouse[4], Diederik van de Beek[5], Anne Hege Aamodt[6,7], Alfonso Ciccone[8], Janika Kõrv[9], Iwona Kurkowska-Jastrzebska[10], Milani Deb-Chatterji[11], Jesse Dawson[12], George Ntaios[13], Götz Thomalla[14], H Bart van der Worp[1], for the PRECIOUS investigators[¶]

1 Department of Neurology and Neurosurgery, Brain Center, University Medical Center Utrecht, Utrecht, Netherlands, 2 Department of Neurology, St. Antonius Hospital, Nieuwegein/Utrecht, The Netherlands, 3 Department of Neurology, Spaarne Gasthuis, Haarlem, The Netherlands, 4 Stroke Trials Unit, University of Nottingham, Nottingham, United Kingdom, 5 Amsterdam UMC, location University of Amsterdam, Department of Neurology, Amsterdam Neuroscience, Amsterdam, The Netherlands, 6 Department of Neurology, Oslo University Hospital, Oslo, Norway, 7 Norway and Department of Neuromedicine and Movement science, Norwegian University of Science and Technology, Trondheim, Norway, 8 Department of Neurology and Stroke Unit, ASST di Mantova, Mantua, Italy, 9 Department of Neurology and Neurosurgery, University of Tartu, Tartu, Estonia, 10 2nd Department of Neurology, Institute of Psychiatry and Neurology, Warsaw, Poland, 11 Department of Neurology, Neurovascular Center, University Hospital Schleswig-Holstein Campus Kiel, Kiel, Germany, 12 Institute of Cardiovascular and Medical Sciences, University of Glasgow, Glasgow, United Kingdom, 13 First Propedeutic Department of Internal Medicine, Aristotle University of Thessaloniki, Thessaloniki, Greece, 14 Department of Neurology, University Medical Center Hamburg-Eppendorf, Hamburg, Germany

¶ PRECIOUS investigators: Please see the appendix of the publication of the main results for a list of the PRECIOUS investigators.
* j.de.jonge@antoniusziekenhuis.nl

## Abstract

### Background and aims

Prophylactic administration of paracetamol (acetaminophen) has been reported to reduce body temperature in the first day after stroke and to reduce blood pressure on the first day. We aimed to validate these findings in the randomised PRECIOUS trial and to assess the effect of paracetamol on body temperature in the first seven days after stroke.

### Patients and methods

PRECIOUS was an international, 3*2 factorial, randomised, controlled, clinical trial assessing preventive treatment with paracetamol, metoclopramide, and ceftriaxone in patients aged 66 years or older with acute stroke and a score on the National Institutes of Health Stroke Scale (NIHSS) ≥ 6. Paracetamol was given in a dose of 4g daily for four days. Vital signs were recorded at 12 hours intervals up to seven days.

**Data availability statement:** Unfortunately, we cannot make data publicly available because of strict privacy regulations and the fact that patients did not explicitly provide consent for public disclosure of their coded data. In the informed consent form, that was signed by the participants and approved by the central ethical committee, it was not mentioned that the data could be made publicly available. We also discussed the situation with the privacy officer of the UMC Utrecht and he agrees with the statement above. The email address that can be used for future references regarding data availability is the general email address of the Scientific Bureau of the Neurology Trial Department at UMC: trialbur@umcutrecht.nl.

**Funding:** PRECIOUS was funded by the European Union's Horizon 2020 research and innovation programme (634809).

**Competing interests:** JCdJ, WMS, HR report grants from the European Union, all paid to their institution. PMB is Stroke Association Professor of Stroke Medicine and an emeritus NIHR Senior Investigator; he has received grants from the NIHR and British Heart Foundation, and funding for consultancy from CoMind and DiaMedica. DvdB reports having received research grants from the European Union, The Netherlands for Health Research and Development, ItsMe Foundation, AMC Foundation and Roche; none related. AHA reports research grants from Boehringer Ingelheim, lectures fee from Abbvie, BMS/Pfizer, Novartis, Roche and Teva and participation in Advisory Board for Lundbeck, Abbvie and MSD; none related. AC reports grants from the European Union and Lombardy region, for research paid to his institution, and fees as consultant or lecturer from Alexxion Pharma, Daiichi Sanky, and Italfarmaco. JK reports lecturer fees from Boehringer Ingelheim, Pfizer and Servier, and travel grants from Boehringer Ingelheim and Servier, none related. MDC has received research grants from the Werner Otto Foundation, speakers honoraria from AstraZeneca and was funded by the Faculty of Medicine, University of Kiel, Germany through its Advanced Clinician Scientist Programme, all outside the submitted work. IKJ reports lectures and travel grants

The primary outcomes of this study were body temperature and systolic and diastolic blood pressure at 24 hours after randomisation, analysed with linear regression. The presence of a subfebrile temperature or fever at 24 hours was a secondary outcome.

## Results

We used data from 1419 of 1493 patients included in PRECIOUS. Paracetamol reduced mean body temperature by 0.1°C (95% CI, 0.0–0.2) at 24 hours after randomisation but did not lower systolic or diastolic blood pressure. Paracetamol reduced the occurrence of subfebrile temperatures or fever at 24 hours from 15.8% to 8.3% ($p < 0.001$). The effects of paracetamol on body temperature were consistent over the four days of treatment.

## Conclusion

Prophylactic use of paracetamol resulted in a modest reduction in mean body temperature and almost halved the rate of subfebrile temperatures or fever at 24 hours after stroke, but had no effect on blood pressure.

---

## Introduction

Subfebrile temperatures and fever in the first days after stroke occur in about one third of patients and have been associated with poorer functional outcomes [1,2]. Data from animal studies suggest that the relation between a higher body temperature and a poorer outcome is causal at least in part [3]. Paracetamol (acetaminophen) is an antipyretic and analgesic drug of which the mechanism of action is probably through inhibition of cyclooxygenase (COX)-1 and COX-2, resulting in a reduction of central nervous system prostaglandin production. Although prophylactic treatment with high-dose paracetamol (6 gram per day) led to a modest reduction in body temperature and to a halving of the number of patients with a subfebrile temperature or fever at 24 hours in the earlier phase 3 Paracetamol (Acetaminophen) In Stroke (PAIS) trial [4], this was not associated with an improvement of functional outcome in that trial, nor in the more recent PREvention of Complications to Improve functional OUtcome in elderly patients with acute Stroke (PRECIOUS) trial [4,5]. Information on the effect of paracetamol on body temperature in stroke patients from previous randomized studies is mainly available for the first 24 hours. These studies reported a reduction by 0.22–0.40°C in patients treated with prophylactic paracetamol as compared to controls [4,6–8]. Information of the effect of paracetamol on body temperature at standardized time intervals during a longer time period after stroke could offer valuable insights into why randomized controlled trials of fever prevention with paracetamol conducted to date have yielded neutral results with regard to long-term functional outcome.

In addition to an elevated body temperature, an increased heart rate and a high blood pressure early after stroke have also been associated with a recurrence of

from Merck, Novartis and Roche, Angelini Pharma and travel grants from Boehringer Ingelheim, none related. GT report grants from the European Union, German Research Foundation, German Federal Ministry of Education and Research, German Innovation Fund for research paid to his institution, and fees as consultant or lecturer from Acandis, Alexion, Amarin, Bayer, Boehringer Ingelheim, Daiichi Sanky, BristolMyersSqibb/Pfizer, and Stryker. HBvdW reports having received grants from the European Union, the Dutch Heart Foundation, and Stryker for research, and funding for consultancy from Bayer, Boehringer Ingelheim, and TargED, all paid to his institution.

cardiovascular events, poor functional outcome and/or mortality [9,10]. In PAIS, high-dose paracetamol was associated with a reduction of systolic blood pressure by 4.5 mmHg at 12 hours from start of treatment, but this effect was no longer present at 24 and 48 hours [11]. In the intensive care population, there is increasing evidence of paracetamol-induced arterial hypotension [12–15]. In contrast, treatment with paracetamol for two weeks has been associated with an increase in mean daytime systolic blood pressure of 4.7 mmHg in patients with hypertension [16]. However, next to PAIS, no data is available on the effect of paracetamol on blood pressure in stroke patients.

In order to gain a better understanding of the neutral findings of the PRECIOUS trial, the objective of this study was to investigate the effect of prophylactic paracetamol on body temperature, blood pressure and heart rate in the first 7 days after symptom onset in patients with acute ischaemic stroke or intracerebral haemorrhage included in PRECIOUS.

## Materials and methods

### Patient selection

We used data from the PRECIOUS trial [5]. This was an investigator-initiated, multi-centre, multi-factorial, open-label, randomised controlled trial with blinded outcome assessment of the preventive use of metoclopramide vs. no metoclopramide, ceftriaxone vs. no ceftriaxone, and paracetamol vs. no paracetamol, in patients aged 66 years or older with moderately severe to severe acute ischaemic stroke or intracerebral haemorrhage, defined as a score on the National Institutes of Health Stroke Scale (NIHSS) of 6 or higher. The study was performed across 82 academic and non-academic hospitals in nine European countries (Estonia, Germany, Greece, Hungary, Italy, the Netherlands, Norway, Poland, United Kingdom). Patients were excluded if they had an active infection requiring antibiotic treatment or if they had a pre-stroke score on the modified Rankin Scale (mRS) of 4 or higher. The protocol, statistical analysis plan and main results have been published earlier [5,17,18]. The trial was approved by the central medical ethics committee of the University Medical Center Utrecht on 3 February 2016 and by national or local research ethics committees in all participating countries. Patients, their legal representatives or independent physicians provided written informed consent for participation in the PRECIOUS trial. The PRECIOUS trial included patients from 1 April 2016 up and until 30 June 2022. Data for this substudy was accessed in the second half of 2024. PRECIOUS is registered (ISRCTN82217627).

### Treatment

Patients were randomly allocated (1:1) to metoclopramide (oral, rectal, or intravenous; 10 mg thrice daily) or no metoclopramide, ceftriaxone (intravenous; 2000 mg once daily) or no ceftriaxone, and paracetamol (oral, rectal, or intravenous; 1000 mg four times daily) or no paracetamol, started within 24 h after symptom onset and continued for four days or until discharge, if earlier. Treatment allocation was open and

based on minimisation through a web-based allocation service. Investigators had the opportunity to censor a single randomisation stratum before randomisation in a specific patient (for example in case of an allergy or a clinical indication for one of the study drugs) or for all patients at their study site (for example because of concerns about the prophylactic use of ceftriaxone). In these cases, randomisation was limited to the other two stratums. For example, if a patient was allergic to ceftriaxone, the ceftriaxone stratum was censored, and randomisation was limited to metoclopramide and paracetamol. All participants received standard of care as determined by each site, including reperfusion therapies. Treating physicians were allowed to start any anti-emetic, antibiotic, or antipyretic drug in patients in any treatment group if clinically indicated. Antipyretic drugs other than paracetamol (e.g., non-steroid anti-inflammatory drugs, aspirin in a dose of more than 300 mg) were not included in the current analysis.

## Study parameters and outcomes

In PRECIOUS, body temperature, blood pressure, and heart rate were registered at baseline (last assessment before randomisation) and, if assessed as part of routine clinical care, every 12 hours (± 3 hours) after randomisation during the first seven days (i.e., 14 intervals up and until 168 hours), or until discharge, if earlier. Measurements were performed according to local protocols, and the method of temperature measurement (tympanic, rectally) was recorded. Patients were included in the present study if they had a measurement at baseline and at any of the time points during follow-up. The primary outcomes of the present study were body temperature and systolic and diastolic blood pressure at 24 hours after randomisation. Secondary outcomes were body temperature and systolic and diastolic blood pressure at the other timepoints during follow-up, and subfebrile temperature or fever (≥37.5°C) and fever (≥38.0°C).

We collected data on baseline characteristics, including age, sex, stroke severity (assessed with the NIHSS), medical history (diabetes mellitus, hypertension), stroke subtype (ischaemic stroke, intracerebral haemorrhage), and treatment with intravenous thrombolysis or endovascular thrombectomy.

## Statistical analysis

Absolute mean differences (with 95% confidence intervals (CI)) in body temperature, heart rate, and systolic and diastolic blood pressure between patients randomised to paracetamol versus no paracetamol, were assessed by multiple linear regression for each 12-hour time point after randomisation in the first 7 days after randomisation. Analyses were adjusted for age, sex, baseline body temperature or blood pressure, stroke type, stroke severity, diabetes mellitus, hypertension, atrial fibrillation, pre-stroke modified Rankin Scale (mRS) score, country, time from stroke to trial treatment, allocation to the other treatment strata (metoclopramide, ceftriaxone), and treatment with intravenous thrombolysis or endovascular treatment. A separate logistic regression for each time point was performed, rather than using a repeated measures approach (e.g., linear mixed model), because our primary interest was to assess the effect of the medication at each individual time point separately, rather than to estimate the overall longitudinal effect. Differences in the rate of subfebrile temperatures or fever between patients randomised to paracetamol and patients not randomised to paracetamol were assessed with a chi-square test. No correction for multiple testing was applied, as the analyses were considered exploratory and hypothesis-generating. In addition, the analyses at each time point are not independent, given that they are repeated measurements within the same individuals over time. A $p$ – value of < 0.05 was considered statically significant. All data were analysed using SPSS version 26.0 (IBM, Armonk, NY).

## Results

A total of 1493 patients were included in PRECIOUS. After excluding one patient whose informed consent form was lost, five patients who withdrew consent immediately after randomisation, 20 patients who had no body temperature or blood pressure measurements after the baseline measurements in the first 7 days after randomisation, and 48 patients who had paracetamol censored during randomisation, 1419 patients were included in the present study, of whom 692 were

randomised to paracetamol and 727 to no paracetamol (see Table 1 for baseline characteristics). Of patients randomised to paracetamol, 544 (78.6%) received at least 75% of all doses of paracetamol. Of the patients randomised to 'no paracetamol,' 275 (37.8%) received at least one dose of an antipyretic drug in the first 4 days after randomisation.

Paracetamol led to a reduction in mean body temperature at 24 hours of 0.1°C (95% CI, 0.0–0.2), p = 0.008) and a reduction in the percentage of subfebrile temperature or fever (≥37.5°C) from 15.8% to 8.3% (p < 0.001) and fever (≥38.0°C) from 4.8% to 1.2% (p < 0.001). Body temperatures in patients randomised to paracetamol were numerically lower than those in the standard treatment group up to and including day 4, after which the study medication was stopped. This reduction was statistically significant at 12, 24, 48 and 72 hours (Fig 1a; Table 2). The occurrence of a subfebrile body temperature or fever was numerically reduced at every time point during the 96 hours treatment period, which was statistically significant at 24, 60, 72 and 84 hours (Fig 1b; Table 3).

Paracetamol had no effect on systolic or diastolic blood pressure at 24 hours and did not significantly reduce systolic or diastolic blood pressure at any point in time during treatment (Fig 1c; S1 Table). Systolic blood pressure was higher after 60 hours (adjusted mean difference 3.6 mmHg, 95% CI 0.1–7.0) and both systolic and diastolic blood pressure were higher at 96 hours (5.1 mmHg, 95% CI 1.8–8.3 and 2.6 mmHg, 95% CI 0.4–4.9, respectively). Paracetamol led to a very modest numerical reduction in heart rate in the first 96 hours, which was statistically significant at 12 hours after randomisation (−2.2 beats per minute, 95% −4.1, −0.3; S2 Table).

**Table 1. Baseline characteristics of the study population.**

|  | N | All | Paracetamol | |
|---|---|---|---|---|
|  |  |  | Yes | None |
| Patients randomized |  | 1419 | 692 | 727 |
| Age | 1419 | 79.8 (7.7) | 79.7 (7.8) | 79.9 (7.7) |
| Sex, male (%) | 1419 | 715 (50.4%) | 351 (50.7%) | 364 (50.1%) |
| Pre-stroke mRS, median [IQR] | 1419 | 0 [0, 2] | 0 [0, 2] | 0 [0, 2] |
| **Medical history** |  |  |  |  |
| Atrial fibrillation (%) | 1419 | 418 (29.5%) | 212 (30.6%) | 206 (28.3%) |
| Hypercholesteremia (%) | 1419 | 510 (35.9%) | 261 (37.7%) | 249 (34.3%) |
| Hypertension (%) | 1419 | 1035 (72.9%) | 490 (70.8%) | 545 (75.0%) |
| Diabetes mellitus (%) | 1419 | 302 (21.3%) | 152 (22.0%) | 150 (20.6%) |
| Time from onset to randomization (minutes) | 1415 | 876 [466, 1230] | 885 [473, 1243] | 867 [458, 1214] |
| **Stroke type** | 1419 |  |  |  |
| Ischemic stroke (%) |  | 1210 (85.3%) | 573 (82.8%) | 637 (87.6%) |
| Intracerebral hemorrhage (%) |  | 190 (13.4%) | 110 (15.9%) | 80 (11.0%) |
| Other diagnosis (%) |  | 19 (1.3%) | 9 (1.3%) | 10 (1.4%) |
| NIHSS total score (/42) | 1419 | 11 [8, 17] | 11 [8, 17] | 12 [8, 17] |
| **Vital signs** |  |  |  |  |
| Systolic blood pressure (mmHg) | 1413 | 152.8 (26.0) | 151.7 (26.5) | 153.9 (25.5) |
| Diastolic blood pressure (mmHg) | 1413 | 80.6 (16.7) | 80.5 (16.7) | 80.7 (16.8) |
| Heart rate (bpm) | 1400 | 78.0 (17.8) | 78.0 (18.1) | 77.9 (17.6) |
| Temperature (°C) | 1314 | 36.5 (0.5) | 36.5 (0.5) | 36.5 (0.5) |
| **Acute stroke treatment**<br>In patients with ischemic stroke |  |  |  |  |
| Intravenous thrombolysis (%) | 1232 | 570 (46.3%) | 280 (47.9%) | 290 (44.8%) |
| Mechanical thrombectomy (%) | 1232 | 292 (23.7%) | 134 (22.9%) | 158 (24.4%) |

Baseline table. Values depicted in mean (SD), median [IQR] or percentages (%). N = number, bpm = beats per minute, IQR = interquartile range.

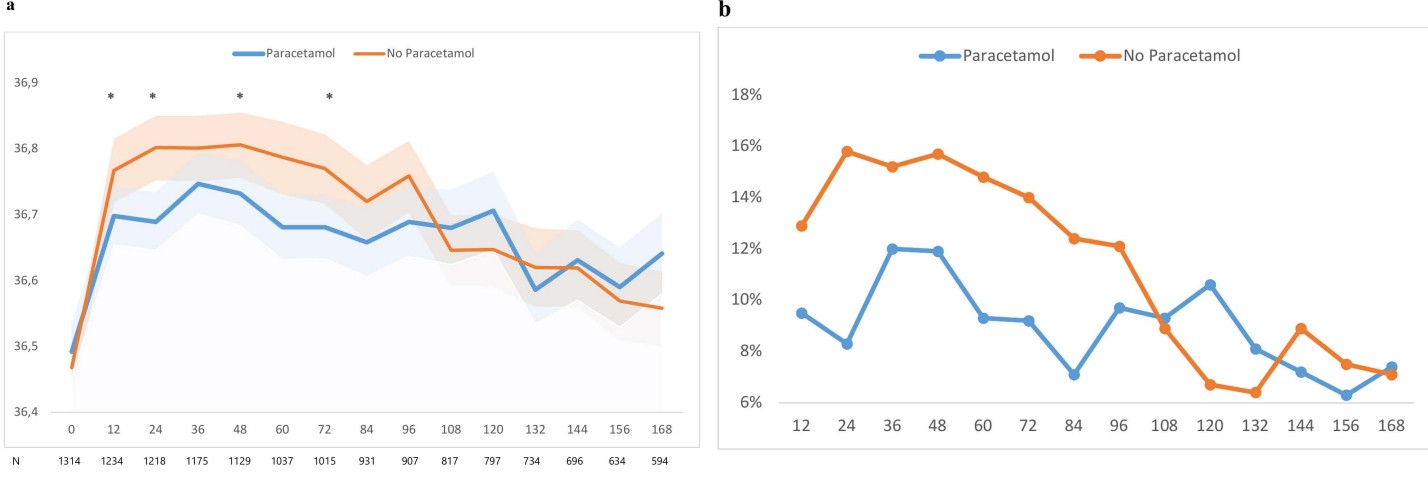

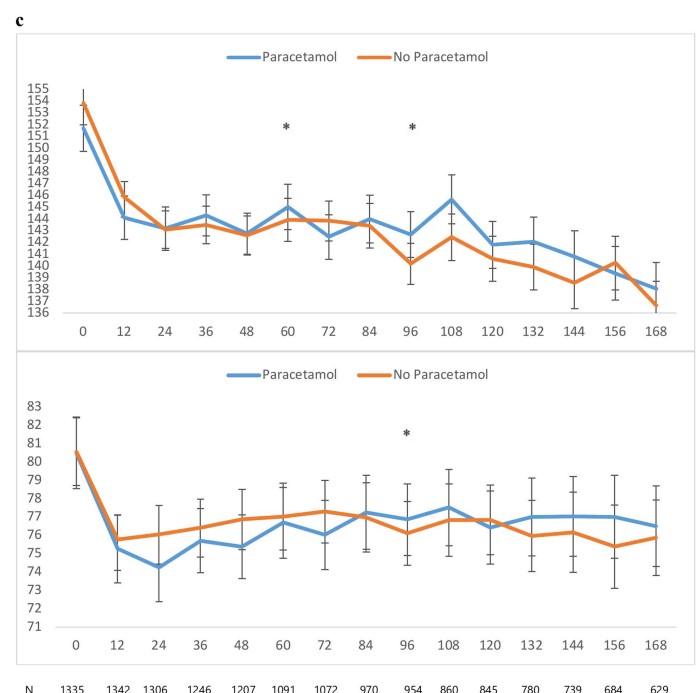

**Fig 1.** **a**. Body temperature and paracetamol: the course of body temperature in the first 7 days after randomisation. The solid lines represent the mean temperatures and the blue and red bands the 95% confidence intervals. *: Statistically significant difference in body temperature between paracetamol and control group after multivariable adjustment. N = number of patients at each time point after randomisation. **b**. Fever and paracetamol: the percentage of patients with a body temperature of ≥37.5°C in the first 7 days after randomisation in the paracetamol and no paracetamol group. *: Statistically significant difference in proportion of patients with temperature ≥ 37.5°C between paracetamol and control group. Numbers of patients at each time point are identical to figure 1a. **c**. Blood pressure and paracetamol: the course of systolic (upper figure) and diastolic (lower figure) blood pressure in the first 7 days after randomisation of patients in the paracetamol and no paracetamol group. *: Statistically significant difference in blood pressure between paracetamol and control group after multivariable adjustment. N = number of patients at each time point after randomisation.

## Discussion

In elderly patients with acute ischaemic stroke or intracerebral haemorrhage included in PRECIOUS, prophylactic treatment with paracetamol reduced the mean body temperature by a very modest 0.1°C at 24 hours after randomisation. This

**Table 2. Mean body temperature in patients with and without paracetamol.**

| Time | Body temperature | | | | | | |
|------|------------------|---|---|---|---|---|---|
| | No paracetamol | | Paracetamol | | | | |
| | Mean (SD) | N | Mean (SD) | N | DIM (95% CI) | Adjusted mean difference# | p-value |
| 0 | 36.5 (0.5) | 639 | 36.5 (0.5) | 675 | 0.0 (0.0 - 0.1) | – | – |
| 12h | 36.8 (0.6) | 599 | 36.7 (0.6) | 635 | −0.1 (−0.1 - 0.0) | −0.1 (−0.2–0.0) | 0.04* |
| 24h | 36.8 (0.6) | 587 | 36.7 (0.5) | 631 | −0.1 (−0.2 - −0.1) | −0.1 (−0.2–0.0) | <0.01* |
| 36h | 36.8 (0.6) | 565 | 36.8 (0.6) | 610 | −0.1 (−0.1 - 0.0) | −0.1 (−0.2–0.0) | 0.76 |
| 48h | 36.8 (0.6) | 544 | 36.7 (0.6) | 585 | −0.1 (−0.1 - 0.0) | −0.1 (−0.2 - 0.0) | 0.04* |
| 60h | 36.8 (0.6) | 503 | 36.7 (0.6) | 534 | −0.1 (−0.2 - 0.0) | −0.1 (−0.2–0.0) | 0.07 |
| 72h | 36.8 (0.6) | 488 | 36.7 (0.5) | 527 | −0.1 (−0.2 - 0.0) | −0.1 (−0.2 - 0.0) | 0.03* |
| 84h | 36.7 (0.6) | 449 | 36.7 (0.6) | 482 | −0.1 (−0.1 - 0.0) | −0.1 (−0.2–0.0) | 0.17 |
| 96h | 36.8 (0.6) | 444 | 36.7 (0.6) | 461 | −0.1 (−0.2 - 0.0) | −0.1 (−0.2–0.0) | 0.08 |
| 108h | 36.7 (0.6) | 399 | 36.7 (0.6) | 418 | 0.0 (0.0 - 0.1) | 0.0 (−0.1–0.1) | 0.56 |
| 120h | 36.7 (0.6) | 395 | 36.7 (0.6) | 402 | 0.1 (−0.0 - 0.1) | 0.1 (−0.1–0.3) | 0.19 |
| 132h | 36.6 (0.6) | 359 | 36.6 (0.5) | 375 | 0.0 (−0.1 - 0.1) | 0.0 (−0.1–0.1) | 0.62 |
| 144h | 36.6 (0.5) | 346 | 36.6 (0.6) | 350 | 0.0 (−0.1 - 0.1) | 0.0 (−0.1–0.1) | 0.93 |
| 156h | 36.6 (0.5) | 316 | 36.6 (0.5) | 318 | 0.0 (−0.1 - 0.1) | 0.0 (−0.1–0.2) | 0.49 |
| 168h | 36.6 (0.5) | 298 | 36.6 (0.5) | 296 | 0.1 (0.0 - 0.2) | 0.1 (−0.1–0.2) | 0.34 |

The mean body temperature of patients randomised to paracetamol or to no paracetamol. Values depicted in mean (standard deviation). DIM: difference in means; h = hours; SD = standard deviation; 95% CI = 95% confidence interval. * = statistically significant. # Analyses were adjusted for age, sex, baseline body temperature, stroke type, stroke severity, diabetes mellitus, hypertension, atrial fibrillation, pre-stroke modified Rankin Scale (mRS) score, baseline body temperature, country, time from stroke to trial treatment, allocation to the other treatment strata (metoclopramide, ceftriaxone), and treatment with intravenous thrombolysis or endovascular treatment.

**Table 3. Rate of subfebrile and febrile body temperature in patients with and without paracetamol prophylaxis.**

| Time | Body temperature | | | | |
|------|------------------|---|---|---|---|
| | No paracetamol | | Paracetamol | | |
| | ≥37.5°C | N | ≥37.5°C | N | p-value |
| 0 | 25 (3.7%) | 675 | 25 (3.9%) | 639 | NA |
| 12h | 82 (12.9%) | 635 | 57 (9.5%) | 599 | 0.06 |
| 24h | 100 (15.8%) | 631 | 49 (8.3%) | 587 | <0.01* |
| 36h | 93 (15.2%) | 610 | 68 (12%) | 565 | 0.11 |
| 48h | 92 (15.7%) | 585 | 65 (11.9%) | 544 | 0.07 |
| 60h | 79 (14.8%) | 534 | 47 (9.3%) | 503 | <0.01* |
| 72h | 74 (14%) | 527 | 45 (9.2%) | 488 | 0.02* |
| 84h | 60 (12.4%) | 482 | 32 (7.1%) | 449 | <0.01* |
| 96h | 56 (12.1%) | 461 | 43 (9.7%) | 444 | 0.25 |
| 108h | 37 (8.9%) | 418 | 37 (9.3%) | 399 | 0.83 |
| 120h | 27 (6.7%) | 402 | 42 (10.6%) | 395 | 0.05 |
| 132h | 24 (6.4%) | 375 | 29 (8.1%) | 359 | 0.38 |
| 144h | 31 (8.9%) | 350 | 25 (7.2%) | 346 | 0.43 |
| 156h | 24 (7.5%) | 318 | 20 (6.3%) | 316 | 0.55 |
| 168h | 21 (7.1%) | 296 | 22 (7.4%) | 298 | 0.89 |

The rate of subfebrile temperature or fever (≥37.5°C) in patients randomised to paracetamol or no paracetamol. Values depicted in percentages (%). * = Statistically significant. h = hours

effect was consistent over the four days or active treatment. Paracetamol also almost halved the occurrence of a subfebrile temperature or fever at 24 hours and reduced the occurrence of fever (≥38.0°C) at this timepoint by about three quarters. In contrast to a previous randomised trial of paracetamol in acute stroke [4,11], paracetamol did not lower systolic or diastolic blood pressure and the effect on heart rate was very small.

Higher body temperatures after stroke have consistently been associated with a poor outcome, both in animal studies [3] and in stroke patients [2,19]. It has previously been estimated that the risk of a poor outcome more than doubles with each 1°C increase in body temperature [20]. However, randomised clinical trials did not show benefit of pharmacological fever prevention in the first days after stroke on functional outcome at 90 days [4,5]. If temperature reduction improves functional outcomes after stroke, the effect, if any, is likely dependent on the extent of the temperature reduction and on body temperatures in the control group. The very modest reduction in mean body temperature of 0.1°C in PRECIOUS and the mean body temperatures ranging from 36.5°C to 36.8°C during the first four days in the standard treatment group reduced the chance of benefit from treatment with paracetamol. In previous phase II and III trials on the effect of paracetamol in patients with acute stroke, the reduction in mean body temperature at 24 hours was more pronounced (0.2 to 0.4°C) [4,6–8]. This may be explained by the higher daily dose of paracetamol (6g versus 4g) and the higher mean temperatures (e.g., 37.1°C at 24 hours in controls in PAIS and 36.7°C in PRECIOUS) in these studies as compared to PRECIOUS [4–7]. The proportion of patients in the standard treatment group with a subfebrile temperature or fever at 24 hours was also considerably higher in PAIS than in PRECIOUS: 30% vs. 16%, with a reduction in the paracetamol group by half in both trials. In PAIS, paracetamol improved functional outcome in patients with a baseline body temperature of 37.0 to 39.0°C, but in PRECIOUS this subgroup was too small for an informative analysis [4].

PRECIOUS included only patients aged 66 years or older because of their greater risk of pneumonia than younger patients [21].The mean age in this trial was 79.8 years. However, in general, older patients are less likely to develop fever. It has been estimated that 20–30% of elderly persons with an infection have a reduced or absent fever response [22].This probably explains the lower temperatures in the first few days in PRECIOUS as compared with PAIS, in which the mean age was 69.8 years [4].

In PRECIOUS, the potential of paracetamol to reduce body temperature was possibly also reduced by two other factors. Firstly, not all of the patients randomised to paracetamol received all 16 gifts. Secondly, clinicians were allowed to treat patients in the standard treatment group with antipyretic medication if clinically indicated. More than one third of the patients who were randomised to no paracetamol received any antipyretic drug in the first four days after randomisation, probably diluting a treatment effect of paracetamol.

A previous post-hoc analysis of the PAIS trial showed that treatment with paracetamol resulted in a lower systolic blood pressure after 12 hours, but not after 24 or 48 hours [11]. It was hypothesised that paracetamol reduces blood pressure by means of COX inhibition or pain relief. In PRECIOUS, we could not replicate this effect of paracetamol on diastolic or systolic blood pressure. As mentioned above, a higher dose of paracetamol was used in PAIS, which could have had a more profound effect on blood pressure. In studies performed in the intensive care unit, it has consistently been found that intravenous paracetamol is associated with clinically relevant episodes of arterial hypotension [23]. The effect is seen within minutes after administration, with a maximum after 15–30 minutes, and a normalisation after approximately one hour. In our study, we do not have data about the time between each study drug administration and blood pressure measurement. Paracetamol was given every 6 hours, and vital signs were recorded every 12 hours (±3 hours). It is therefore possible that an effect of paracetamol on blood pressure was missed due to the lack of standardised strict timing of the assessment of vital signs. Nonetheless, we found no evidence that prophylactic treatment with paracetamol after stroke has any clinically relevant effect on blood pressure.

In conclusion, prophylactic treatment with paracetamol led to a decrease in body temperature at 24 hours after stroke and a numerical decrease during the four day treatment period. The decrease of 0.1°C was very small, however, and

less pronounced than previously reported in literature. Prophylactic treatment with paracetamol reduced the number of patients with subfebrile or febrile body temperature after 24 hours by half. In the PRECIOUS study, paracetamol did not lower systolic or diastolic blood pressure. In this population, prophylactic administration of paracetamol therefore has no added benefit to regular stroke unit care, which includes the administration of antipyretic medication if clinically indicated.

## Supporting information

**S1 Table. Mean systolic and diastolic blood pressure in patients with and without paracetamol.**
(DOCX)

**S2 Table. Mean heart rate in patients with and without paracetamol.**
(DOCX)

**S3 Appendix. PRECIOUS investigators.**
(DOCX)

## Acknowledgments

We thank all patients and their representatives for their participation in the trial, and all local investigators for their contributions.

## Author contributions

**Conceptualization:** Jeroen C. de Jonge, Wouter M. Sluis, Hendrik Reinink, Philip Bath, Diederik van de Beek, Anne Hege Aamodt, Alfonso Ciccone, Janika Korv, Iwona Kurkowska-Jastrzebska, George Ntaios, Gotz Thomalla, H Bart van der Worp.

**Data curation:** Jeroen C. de Jonge, Wouter M. Sluis, Hendrik Reinink, Lisa J Woodhouse, H Bart van der Worp.

**Formal analysis:** Jeroen C. de Jonge, Hendrik Reinink, Philip Bath, Lisa J Woodhouse.

**Funding acquisition:** H Bart van der Worp.

**Investigation:** Jeroen C. de Jonge, Philip Bath, Diederik van de Beek, Anne Hege Aamodt, Alfonso Ciccone, Janika Korv, Iwona Kurkowska-Jastrzebska, George Ntaios, Gotz Thomalla, H Bart van der Worp.

**Methodology:** Jeroen C. de Jonge, Philip Bath, Lisa J Woodhouse, H Bart van der Worp.

**Project administration:** Jeroen C. de Jonge, Philip Bath, Diederik van de Beek, Anne Hege Aamodt, Alfonso Ciccone, Janika Korv, Iwona Kurkowska-Jastrzebska, Milani Deb-Chatterji, Jesse Dawson, George Ntaios, Gotz Thomalla, H Bart van der Worp.

**Resources:** H Bart van der Worp.

**Software:** Jeroen C. de Jonge.

**Supervision:** Wouter M. Sluis, H Bart van der Worp.

**Validation:** H Bart van der Worp.

**Visualization:** H Bart van der Worp.

**Writing – original draft:** Jeroen C. de Jonge.

**Writing – review & editing:** Wouter M. Sluis, Hendrik Reinink, Philip Bath, Lisa J Woodhouse, Diederik van de Beek, Anne Hege Aamodt, Alfonso Ciccone, Janika Korv, Iwona Kurkowska-Jastrzebska, Milani Deb-Chatterji, Jesse Dawson, George Ntaios, Gotz Thomalla, H Bart van der Worp.

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
