## [Decision Letter · Decision Letter 0]

23 Jun 2025

PONE-D-25-22053
The effects of paracetamol on body temperature and blood pressure in patients with acute stroke. Data from the PRECIOUS trial.
PLOS ONE

Dear Dr. de Jonge,

Thank you for submitting your manuscript to PLOS ONE. After careful consideration, we feel that it has merit but does not fully meet PLOS ONE’s publication criteria as it currently stands. Therefore, we invite you to submit a revised version of the manuscript that addresses the points raised during the review process.

**ACADEMIC EDITOR:**

Heartfelt thanks to the authors for submitting their work to PLOS ONE for consideration for publications. A number of critical areas have been identified in the manuscript that require revision to improve its scholarly content and quality. These include critical issues in specific areas identified by the reviewers including introduction, methodology, results, discussion and conclusion sections. Furthermore, authors should also address the key concerns regarding the choice of statistical methods, and credibility of the statistical analysis, interpretation and inferential judgement presented in the manuscript. Furthermore, the questions raised regarding the weak justification and lack of knowledge gaps should also be properly addressed in the revised manuscript. Lastly, authors should revise the whole manuscript for grammar and syntax to improve readability and sharpen its focus. Best wishes and I look forward to seeing your revised manuscript with an improved scholarly quality and content.

We look forward to receiving your revised manuscript.

Kind regards,

Kazeem Babatunde Yusuff, Ph.D

Academic Editor

PLOS ONE

Journal Requirements:

[PRECIOUS was funded by the European Union’s Horizon 2020 research and innovation programme (634809).].

4. In the online submission form, you indicated that [De-identified individual participant data and a data dictionary defining each field in the set can be made available to others upon reasonable request to the corresponding author, subject to privacy regulation.].

6. Thank you for stating the following in the Competing Interests section:

[JCdJ, WMS, HR report grants from the European Union, all paid to their institution. PMB is Stroke Association Professor of Stroke Medicine and an emeritus NIHR Senior Investigator; he has received grants from the NIHR and British Heart Foundation, and funding for consultancy from CoMind and DiaMedica. DvdB reports having received research grants from the European Union, The Netherlands for Health Research and Development, ItsMe Foundation, AMC Foundation and Roche; none related. AHA reports research grants from Boehringer Ingelheim, lectures fee from Abbvie, BMS/Pfizer, Novartis, Roche and Teva and participation in Advisory Board for Lundbeck, Abbvie and MSD; none related. AC reports grants from the European Union and Lombardy region, for research paid to his institution, and fees as consultant or lecturer from Alexxion Pharma, Daiichi Sanky, and Italfarmaco. JK reports lecturer fees from Boehringer Ingelheim, Pfizer and Servier, and travel grants from Boehringer Ingelheim and Servier, none related. MDC has received research grants from the Werner Otto Foundation, speakers honoraria from AstraZeneca and was funded by the Faculty of Medicine, University of Kiel, Germany through its Advanced Clinician Scientist Programme, all outside the submitted work. IKJ reports lectures and travel grants from Merck, Novartis and Roche, Angelini Pharma and travel grants from Boehringer Ingelheim, none related. GT report grants from the European Union, German Research Foundation, German Federal Ministry of Education and Research, German Innovation Fund for research paid to his institution, and fees as consultant or lecturer from Acandis, Alexion, Amarin, Bayer, Boehringer Ingelheim, Daiichi Sanky, BristolMyersSqibb/Pfizer, and Stryker. HBvdW reports having received grants from the European Union, the Dutch Heart Foundation, and Stryker for research, and funding for consultancy from Bayer, Boehringer Ingelheim, and TargED, all paid to his institution.].

We note that you received funding from a commercial source: [European Union, NIHR, British Heart Foundation, Abbvie, BMS/Pfizer, Lombardy region, Alexxion Pharma, Daiichi Sanky, Italfarmaco, Boehringer Ingelheim, Pfizer and Servier, Werner Otto Foundation, Faculty of Medicine, University of Kiel, Merck, Novartis and Roche, Angelini Pharma, German Research Foundation, German Federal Ministry of Education and Research, Acandis, Alexion, Amarin, Bayer, Boehringer Ingelheim, Daiichi Sanky, BristolMyersSqibb/Pfizer, and Stryker]

Within this Competing Interests Statement, please confirm that this does not alter your adherence to all PLOS ONE policies on sharing data and materials by including the following statement: ""This does not alter our adherence to PLOS ONE policies on sharing data and materials.” (as detailed online in our guide for authors http://journals.plos.org/plosone/s/competing-interests).  If there are restrictions on sharing of data and/or materials, please state these. Please note that we cannot proceed with consideration of your article until this information has been declared.

7. One of the noted authors is a group [PRECIOUS investigators]. In addition to naming the author group, please list the individual authors and affiliations within this group in the acknowledgments section of your manuscript. Please also indicate clearly a lead author for this group along with a contact email address.

Reviewers' comments:

Reviewer's Responses to Questions

**Comments to the Author**

1. Is the manuscript technically sound, and do the data support the conclusions?

Reviewer #1: Partly

Reviewer #2: Partly

Reviewer #3: Yes

2. Has the statistical analysis been performed appropriately and rigorously? 

Reviewer #1: No

Reviewer #2: Yes

Reviewer #3: Yes

3. Have the authors made all data underlying the findings in their manuscript fully available?

Reviewer #1: Yes

Reviewer #2: Yes

Reviewer #3: Yes

4. Is the manuscript presented in an intelligible fashion and written in standard English?

Reviewer #1: Yes

Reviewer #2: Yes

Reviewer #3: Yes

5. Review Comments to the Author

Reviewer #1: Reviewer Comments

This manuscript evaluates the effects of prophylactic paracetamol administration on body temperature and blood pressure among elderly acute stroke patients, leveraging data from the international PRECIOUS randomized controlled trial (RCT). The analysis of 1,419 patients aimed to validate previous findings from studies such as the PAIS trial. Your primary conclusion—that paracetamol modestly reduces mean body temperature and significantly decreases the occurrence of subfebrile or febrile episodes without affecting blood pressure—is clinically relevant and clearly articulated.

Overall, the manuscript is logically structured and well-presented. The international, multicenter RCT design ensures robust statistical power and enhances external validity. However, important concerns regarding the accuracy and transparency of your statistical analysis—specifically related to the chi-square test results—need to be urgently addressed, as these issues directly influence the credibility and interpretability of your findings.

Specific Comments:

1. Multiple Comparisons and Potential for Inflated Type I Error:

Given that statistical analyses were conducted across multiple time points (every 12 hours over a 7-day period), there is a significant risk of inflated Type I errors (false positives). The manuscript does not clearly indicate whether any statistical corrections (such as Bonferroni, Holm, or False Discovery Rate adjustments) for multiple comparisons were applied.

Recommendation:

Explicitly clarify whether any correction for multiple comparisons was performed. If corrections were not applied, please provide a clear justification for this decision in the manuscript. Transparently acknowledging this issue will considerably enhance the methodological rigor and reliability of your analysis.

2. Clarification Regarding "Adjusted Difference" in Table 2:

In Table 2, you report an 'Adjusted difference'; however, the methods section lacks a clear explanation of how these adjusted mean differences were derived.

Recommendation:

Clearly explain the statistical approach and the specific covariates used for adjustment when calculating these differences. This should be explicitly detailed in the statistical methods section of your manuscript to ensure clarity and reproducibility.

3. Major Concern – Discrepancies in Chi-square Test Results (Table 3):

In your methods section, you indicated that differences in the rate of subfebrile temperatures or fever were assessed using chi-square tests, with results presented in Table 3. Upon independently recalculating these chi-square values using the provided data, I identified discrepancies between your reported p-values and the independently calculated ones. While these discrepancies generally do not appear large enough to substantially alter your primary conclusions, they nevertheless undermine the robustness and credibility of the statistical analysis performed.

Notably, these differences remained even when employing Fisher’s Exact Test, suggesting a systemic issue rather than simply the choice of statistical test. Furthermore, because the variables utilized in your multiple regression analyses were not fully provided, it was impossible for me to independently validate those analyses as well.

Recommendation:

Please urgently re-examine and confirm the accuracy of all chi-square analyses presented in Table 3 using reliable statistical software (e.g., R, Python, SPSS, Stata). Provide transparent recalculations of these analyses along with clearly documented steps or scripts to facilitate external verification. Clarifying this issue is critical to ensure confidence in the reported results and their interpretation. I have shared with the editorial team the Python script and resulting calculations I performed for further reference and verification.

Addressing these concerns will significantly enhance the methodological clarity, transparency, and overall quality of your manuscript.

Reviewer #2: This was a substudy of the PRECIOUS study. Prophylactic administration of paracetamol (acetaminophen) has been reported to reduce body temperature in the first days after stroke and to reduce blood pressure on the first day. The investigators wanted to validate these findings in the randomized PRECIOUS trial. The paper was well presented from the statistical perspective.

Absolute mean differences (with 95% confidence intervals (CI)) in body temperature, heart rate, and systolic and diastolic blood pressure between patients randomized to paracetamol versus no paracetamol, were assessed by multiple linear regression for each 12-hour time point after randomization in the first 7 days after randomization. Performing this type of timepoint analysis the investigators concluded that prophylactic use of paracetamol resulted in a modest reduction in mean body temperature and almost halved the rate of subfebrile temperatures or fever at 24 hours after stroke, but had no effect on blood pressure.

Figures 1a to 1c basically demonstrated the results. The analysis was by time and obviously was very simple. Although this was not the intent of the original trial and the results are mostly observational, one wonders why the investigators did not attempt a repeated measures type of analysis with treatment and time as the factors and getting a p-value for these overall effects for the course of this short study. Also please comment on the interaction noticed in the plots, mostly at the end of the Figures 1a and 1b.

Also since a multiple regression analysis adjustment was made, was there any attempt to examine the significance of the baseline values on the outcome?

Reviewer #3: Subject: Peer Review Feedback – Manuscript ID: PONE-D-25-22053

Title: The effects of paracetamol on body temperature and blood pressure in patients with acute stroke. Data from the PRECIOUS trial

Thank you for the opportunity to review this manuscript. The submitted work represents a secondary analysis of the PRECIOUS trial - an international, multicenter, 3×2 factorial, randomized, controlled trial investigating the effects of paracetamol, metoclopramide, and ceftriaxone in older adults (≥66 years) with acute stroke and NIHSS score ≥6. The PRECIOUS trial’s main outcomes, protocol, and statistical analysis plan have previously been published in peer-reviewed journals.

This manuscript investigates a clinically relevant topic – i.e. the effects of paracetamol on temperature and blood pressure (BP) in acute stroke patients. While the study addresses an important area with limited existing evidence, the manuscript in its current form requires significant revision to meet the standards of PLOS ONE. Below are my detailed comments and suggestions for improvement.

General Comments

• The manuscript is generally well-structured and presents interesting findings. However, improvements are needed in language, grammar, and scientific writing quality.

• I strongly recommend a thorough proofreading by a native English speaker with experience in clinical research to enhance clarity and coherence.

• The rationale for conducting this secondary analysis is weakly presented. A more robust literature review is needed to clearly articulate the knowledge gap this study addresses.

• All abbreviations should be defined at first mention and used consistently throughout. For example, NIHSS was not defined in the abstract.

• Subheadings in various sections are inconsistently formatted.

Title

• The current title does not fully reflect the study population. Since all included participants were ≥66 years, I recommend revising the title to include “older adult patients” for accuracy.

Abstract

• Background and Aims: The rationale for the study is not clearly stated. Prior studies have reported on paracetamol’s effects in stroke, what new aspect is being investigated here?

• Methods: The outcome descriptions must be consistent with the main text (e.g., diastolic BP is included in the main manuscript but not in the abstract).

• Results: When referring to “blood pressure,” clarify whether this means “systolic blood pressure” specifically.

Introduction

• The introduction should better define what is known, what remains unknown, and the research gap.

• Line 72: Specify what is meant by “functional outcomes.”

• Line 77: Define what is considered a “high dose” of paracetamol and cite relevant sources.

• Line 83–84: Expand on the association between elevated temperature/BP and poor clinical outcomes, including which outcomes.

• Line 87: Replace “by contrast” with “in contrast.”

• Line 90–92: Clarify the objective statement, which is currently confusing due to the introduction of PAIS.

Methods

• Lines 96–104: Please cite the original publication(s) of the PRECIOUS trial. The long sentence here should be broken into shorter, clearer ones.

• Line 109–110: Dates should follow academic and journal-specific formatting.

• Line 113: Ensure uniformity in formatting of subheadings (some are bold, others italicized).

• Line 119–122: Clarify the statement about omitting one randomization stratum.

• Line 136–137: Ensure consistency in the list of primary outcomes across sections.

• Line 151: Avoid redundancy - baseline body temperature was already listed.

• Line 155–156: “Statistical significance was set at 95%” is incorrect. Please revise to “a p-value < 0.05 was considered statistically significant.”

• Line 156: The city of manufacture for SPSS should be verified. Chicago, IL may not be accurate.

Results

• Line 177 onwards (last paragraph) – I see potential reporting bias in the paper. The reporting of systolic BP findings is detailed, but diastolic BP is completely omitted, despite being a listed primary outcome.

• Please present findings on diastolic BP, in the text and/or through an additional table or figure.

Discussion

• The discussion is too descriptive. A more critical interpretation of the findings in the context of existing literature is required.

• Avoid redundancy and focus on key outcomes and their implications.

• Line 188: Reword “by a very modest 0.1°C” to improve clarity and tone.

• Line 191: Please cite the previous randomized trial referenced.

• Line 193–194 and Line 204–205: Address grammatical errors in these sentences.

• Consider including interpretation of findings related to DBP and HR, which are currently underrepresented.

Conclusion

• The conclusion should be more reflective of the study’s findings and emphasize clinical implications. Ensure alignment with the study’s objectives.

Tables

• Table 1:

o Title of Table 1 is not descriptive enough.

o Is the age presented as mean (SD), please specify in the first column?

o Medical history: Some have % in the first column, others like AF and DM do not have the symbol of %.

• Table 2:

o Expand the title for clarity – it is currently not descriptive enough.

o A similar table for diastolic BP should be created and included.

o

References

• Ensure all in-text citations and reference formatting follow PLOS ONE’s guidelines.

Reviewer’s Summary

In summary, the manuscript has the potential to contribute valuable data on a topic of interest in stroke management. However, substantial revision is necessary to clarify its rationale, improve its consistency, and enhance scientific rigor. I hope the authors will find these comments constructive in revising the manuscript.

6. PLOS authors have the option to publish the peer review history of their article (what does this mean?). If published, this will include your full peer review and any attached files.

Reviewer #1: No

Reviewer #2: No

Reviewer #3: No

---

## [Author Response · Author response to Decision Letter 1]

3 Oct 2025

Dear Dr. Yusuff,

Thank you for considering the publication of our paper in your journal and for the constructive criticism of the reviewers. We have read the suggestions for improvement with detail and adjusted the manuscript where appropriate. Please see below our response to each point raised by the academic editor and reviewers.

Journal Requirements:

The paper has been adjusted according to the above-mentioned style requirements.

In the ´Funding Information´ section, it is not possible to select one of the pre-defined options that exactly resembles our standard funding statement: ´This project has received funding from the European Union´s Horizon 2020 research and innovation programme under grant agreement No: 634809. Currently, we have selected the heading ´H2020 European Research Council´, as this option best resembles the funding statement.

 [PRECIOUS was funded by the European Union’s Horizon 2020 research and innovation programme (634809).].

The statement has been added to the revised cover letter.

4. In the online submission form, you indicated that [De-identified individual participant data and a data dictionary defining each field in the set can be made available to others upon reasonable request to the corresponding author, subject to privacy regulation.].

The statement has been adjusted. Unfortunately, we cannot make data publicly available because of strict privacy regulations and the fact that patients did not explicitly provide consent for public disclosure of their coded data. In the informed consent form, that was signed by the participants and approved by the central ethical committee, it was not mentioned that the data could be made publicly available. We also discussed the situation with the privacy officer of the UMC Utrecht and he agrees with the statement above.

The captions for the Supporting Information files have been added to the end of our manuscript.

6. Thank you for stating the following in the Competing Interests section:

[JCdJ, WMS, HR report grants from the European Union, all paid to their institution. PMB is Stroke Association Professor of Stroke Medicine and an emeritus NIHR Senior Investigator; he has received grants from the NIHR and British Heart Foundation, and funding for consultancy from CoMind and DiaMedica. DvdB reports having received research grants from the European Union, The Netherlands for Health Research and Development, ItsMe Foundation, AMC Foundation and Roche; none related. AHA reports research grants from Boehringer Ingelheim, lectures fee from Abbvie, BMS/Pfizer, Novartis, Roche and Teva and participation in Advisory Board for Lundbeck, Abbvie and MSD; none related. AC reports grants from the European Union and Lombardy region, for research paid to his institution, and fees as consultant or lecturer from Alexxion Pharma, Daiichi Sanky, and Italfarmaco. JK reports lecturer fees from Boehringer Ingelheim, Pfizer and Servier, and travel grants from Boehringer Ingelheim and Servier, none related. MDC has received research grants from the Werner Otto Foundation, speakers honoraria from AstraZeneca and was funded by the Faculty of Medicine, University of Kiel, Germany through its Advanced Clinician Scientist Programme, all outside the submitted work. IKJ reports lectures and travel grants from Merck, Novartis and Roche, Angelini Pharma and travel grants from Boehringer Ingelheim, none related. GT report grants from the European Union, German Research Foundation, German Federal Ministry of Education and Research, German Innovation Fund for research paid to his institution, and fees as consultant or lecturer from Acandis, Alexion, Amarin, Bayer, Boehringer Ingelheim, Daiichi Sanky, BristolMyersSqibb/Pfizer, and Stryker. HBvdW reports having received grants from the European Union, the Dutch Heart Foundation, and Stryker for research, and funding for consultancy from Bayer, Boehringer Ingelheim, and TargED, all paid to his institution.].

We note that you received funding from a commercial source: [European Union, NIHR, British Heart Foundation, Abbvie, BMS/Pfizer, Lombardy region, Alexxion Pharma, Daiichi Sanky, Italfarmaco, Boehringer Ingelheim, Pfizer and Servier, Werner Otto Foundation, Faculty of Medicine, University of Kiel, Merck, Novartis and Roche, Angelini Pharma, German Research Foundation, German Federal Ministry of Education and Research, Acandis, Alexion, Amarin, Bayer, Boehringer Ingelheim, Daiichi Sanky, BristolMyersSqibb/Pfizer, and Stryker]

Within this Competing Interests Statement, please confirm that this does not alter your adherence to all PLOS ONE policies on sharing data and materials by including the following statement: ""This does not alter our adherence to PLOS ONE policies on sharing data and materials.” (as detailed online in our guide for authors http://journals.plos.org/plosone/s/competing-interests). If there are restrictions on sharing of data and/or materials, please state these. Please note that we cannot proceed with consideration of your article until this information has been declared.

The amended Competing Interests Statement has been added to our cover letter. Of note, the study itself was only funded by the European Union, which is non-commercial. Individual investigators did have their own competing interests, that were noted in this section.

7. One of the noted authors is a group [PRECIOUS investigators]. In addition to naming the author group, please list the individual authors and affiliations within this group in the acknowledgments section of your manuscript. Please also indicate clearly a lead author for this group along with a contact email address.

Given the length of the list of all the PRECIOUS investigators, the individual authors have been added as supplemental information (S3 appendix).

Reviewers' comments:

Reviewer's Responses to Questions

Comments to the Author

1. Is the manuscript technically sound, and do the data support the conclusions?

Reviewer #1: Partly

Reviewer #2: Partly

Reviewer #3: Yes

2. Has the statistical analysis been performed appropriately and rigorously?

Reviewer #1: No

Reviewer #2: Yes

Reviewer #3: Yes

3. Have the authors made all data underlying the findings in their manuscript fully available?

Reviewer #1: Yes

Reviewer #2: Yes

Reviewer #3: Yes

4. Is the manuscript presented in an intelligible fashion and written in standard English?

Reviewer #1: Yes

Reviewer #2: Yes

Reviewer #3: Yes

5. Review Comments to the Author

Reviewer #1: Reviewer Comments

This manuscript evaluates the effects of prophylactic paracetamol administration on body temperature and blood pressure among elderly acute stroke patients, leveraging data from the international PRECIOUS randomized controlled trial (RCT). The analysis of 1,419 patients aimed to validate previous findings from studies such as the PAIS trial. Your primary conclusion—that paracetamol modestly reduces mean body temperature and significantly decreases the occurrence of subfebrile or febrile episodes without affecting blood pressure—is clinically relevant and clearly articulated.

Overall, the manuscript is logically structured and well-presented. The international, multicenter RCT design ensures robust statistical power and enhances external validity. However, important concerns regarding the accuracy and transparency of your statistical analysis—specifically related to the chi-square test results—need to be urgently addressed, as these issues directly influence the credibility and interpretability of your findings.

Specific Comments:

1. Multiple Comparisons and Potential for Inflated Type I Error:

Given that statistical analyses were conducted across multiple time points (every 12 hours over a 7-day period), there is a significant risk of inflated Type I errors (false positives). The manuscript does not clearly indicate whether any statistical corrections (such as Bonferroni, Holm, or False Discovery Rate adjustments) for multiple comparisons were applied.

Recommendation:

Explicitly clarify whether any correction for multiple comparisons was performed. If corrections were not applied, please provide a clear justification for this decision in the manuscript. Transparently acknowledging this issue will considerably enhance the methodological rigor and reliability of your analysis.

Response:

No corrections for multiple comparisons were applied, as the analyses were considered exploratory and hypothesis-generating rather than confirmatory hypothesis-testing. Furthermore, the multiple measurements were not independent, as the measurements were repeated within the same participants. These two justifications have now been included in the Methods section of the paper. We discussed this with two co-authors of the paper who are also statistician, and they agreed with the statement above.

2. Clarification Regarding "Adjusted Difference" in Table 2:

In Table 2, you report an 'Adjusted difference'; however, the methods section lacks a clear explanation of how these adjusted mean differences were derived.

Recommendation:

Clearly explain the statistical approach and the specific covariates used for adjustment when calculating these differences. This should be explicitly detailed in the statistical methods section of your manuscript to ensure clarity and reproducibility.

Response:

The statistical analyses performed to obtain the results of Table 2 were carried out according to the ´statistical analysis´ paragraph in the ´Methods´ section of the paper. We performed a multiple linear regression and adjusted for the covariates defined in the statistical analysis paragraph: analyses were adjusted for age, sex, baseline body temperature, stroke type, stroke severity, diabetes mellitus, hypertension, atrial fibrillation, pre-stroke modified Rankin Scale (mRS) score, baseline body temperature, country, time from stroke to trial treatment, allocation to the other treatment strata (metoclopramide, ceftriaxone), and treatment with intravenous thrombolysis or endovascular treatment.

The specific covariates have been added to the legend of the Table.

3. Major Concern – Discrepancies in Chi-square Test Results (Table 3):

In your methods section, you indicated that differences in the rate of subfebrile temperatures or fever were assessed using chi-square tests, with results presented in Table 3. Upon independently recalculating these chi-square values using the provided data, I identified discrepancies between your reported p-values and the independently calculated ones. While these discrepancies generally do not appear large enough to substantially alter your primary conclusions, they nevertheless undermine the robustness and credibility of the statistical analysis performed.

Notably, these differences remained even when employing Fisher’s Exact Test, suggesting a systemic issue rather than simply the choice of statistical test. Furthermore, because the variables utilized in your multiple regression analyses were not fully provided, it was impossible for me to independently validate those analyses as well.

Recommendation:

Please urgently re-examine and confirm the accuracy of all chi-square analyses presented in Table 3 using reliable statistical software (e.g., R, Python, SPSS, Stata). Provide transparent recalculations of these analyses along with clearly documented steps or scripts to facilitate external verification. Clarifying this issue is critical to ensure confidence in the reported results and their interpretation. I have shared with the editorial team the Python script and resulting calculations I performed for further reference and verification.

Addressing these concerns will significantly enhance the methodological clarity, transparency, and overall quality of your manuscript.

Response:

The chi-square analyses of Table 3 have been repeated using the same software (SPSS), and the results are identical to the current numbers in Table 3. In the revision, a document containing the exact SPSS output (2x2 contingency tables & chi-square analyses) of each calculation has been added.

It seems that the reviewer applied Yates correction to the chi-square test. We did not use Yates´ correction (or Fisher exact test), because the sample was of considerable size and we felt that Yates´ correction was not appropriate. After discussion with our statistician, we decided not

---

## [Decision Letter · Decision Letter 1]

8 Dec 2025

PONE-D-25-22053R1
The effects of paracetamol on body temperature and blood pressure in elderly patients with acute stroke. Data from the PRECIOUS trial.
PLOS One

Dear Dr. de Jonge,

Thank you for submitting your manuscript to PLOS ONE. After careful consideration, we feel that it has merit but does not fully meet PLOS ONE’s publication criteria as it currently stands. Therefore, we invite you to submit a revised version of the manuscript that addresses the points raised during the review process.

**ACADEMIC EDITOR:**

Heartfelt thanks to the authors for addressing the gaps identified in the previous reviews in the revised manuscript.  However, a few minor revisions have been suggested in the study title and figures.

We look forward to receiving your revised manuscript.

Kind regards,

Kazeem Babatunde Yusuff, Ph.D

Academic Editor

PLOS One

Journal Requirements:

Reviewers' comments:

Reviewer's Responses to Questions

**Comments to the Author**

1. If the authors have adequately addressed your comments raised in a previous round of review and you feel that this manuscript is now acceptable for publication, you may indicate that here to bypass the “Comments to the Author” section, enter your conflict of interest statement in the “Confidential to Editor” section, and submit your "Accept" recommendation.

Reviewer #2: All comments have been addressed

Reviewer #3: (No Response)

2. Is the manuscript technically sound, and do the data support the conclusions?

Reviewer #2: (No Response)

Reviewer #3: Yes

3. Has the statistical analysis been performed appropriately and rigorously? 

Reviewer #2: (No Response)

Reviewer #3: Yes

4. Have the authors made all data underlying the findings in their manuscript fully available?

Reviewer #2: (No Response)

Reviewer #3: Yes

5. Is the manuscript presented in an intelligible fashion and written in standard English?

Reviewer #2: (No Response)

Reviewer #3: Yes

6. Review Comments to the Author

Reviewer #2: (No Response)

Reviewer #3: 1. I suggest the title should include the word "prophylactic" as the management here was not therapeutic. I suggest something like: "The Effects of Prophylactic Paracetamol use on Body Temperature and Blood Pressure in Patients with Acute Stroke".

2. I still feel Figures 1a, 1b, 1c are not of good resolution. Kindly improve the resolution.

7. PLOS authors have the option to publish the peer review history of their article (what does this mean?). If published, this will include your full peer review and any attached files.

Reviewer #2: No

Reviewer #3: No

---

## [Author Response · Author response to Decision Letter 2]

20 Jan 2026

Dear Dr. Yusuff,

Thank you for considering the publication of our paper in your journal and for the constructive criticism of the reviewers. We have read the suggestions for improvement with detail and adjusted the manuscript where appropriate. Please see below our response to each point raised by the academic editor and reviewers.

---

Reviewers' comments:

6. Review Comments to the Author

Reviewer #2: (No Response)

Reviewer #3: 1. I suggest the title should include the word "prophylactic" as the management here was not therapeutic. I suggest something like: "The Effects of Prophylactic Paracetamol use on Body Temperature and Blood Pressure in Patients with Acute Stroke".

2. I still feel Figures 1a, 1b, 1c are not of good resolution. Kindly improve the resolution.

The title has been adapted accordingly.

The resolution of Figure 1B deteriorated during the transfer from Excel to Word. The figure has now been recopied while preserving the original resolution. The resolution of the figures in the Word files appears sufficient at this point. The Word files were subsequently converted to TIFF using an alternative method, resulting in significantly improved image quality. Off note, the quality of the images is low in the overall PDF file of the submission, but if the images are downloaded in the PDF file, the quality is adequate.

---

## [Editor Report · Decision Letter 2]

29 Jan 2026

The effects of prophylactic use of paracetamol on body temperature and blood pressure in elderly patients with acute stroke. Data from the PRECIOUS trial.

PONE-D-25-22053R2

Dear Dr. de Jonge,

We’re pleased to inform you that your manuscript has been judged scientifically suitable for publication and will be formally accepted for publication once it meets all outstanding technical requirements.

Kind regards,

Kazeem Babatunde Yusuff, Ph.D

Academic Editor

PLOS One
---

## [Editor Report · Acceptance letter]

PONE-D-25-22053R2

PLOS One

Dear Dr. de Jonge,

I'm pleased to inform you that your manuscript has been deemed suitable for publication in PLOS One. Congratulations! Your manuscript is now being handed over to our production team.

Kind regards,

on behalf of

Prof. Kazeem Babatunde Yusuff

Academic Editor

PLOS One